# CCN2 Mediates S1P-Induced Upregulation of COX2 Expression in Human Granulosa-Lutein Cells

**DOI:** 10.3390/cells8111445

**Published:** 2019-11-15

**Authors:** Liao-Liao Hu, Hsun-Ming Chang, Yuyin Yi, Yingtao Liu, Elizabeth L. Taylor, Li-Ping Zheng, Peter C.K. Leung

**Affiliations:** 1Jiangxi Medical College, Nanchang University, Nanchang 330031, Jiangxi, China; 2Jiangxi Key Laboratory of Reproductive Physiology and Pathology, Nanchang University, Nanchang 330031, Jiangxi, China; 3Department of Obstetrics and Gynaecology, BC Children’s Hospital Research Institute, University of British Columbia, Vancouver, BC V6H3V5, Canada; CHANGOBS@HOTMAIL.COM (H.-M.C.); yuyinyi0805@gmail.com (Y.Y.); liuyingtao@fudan.edu.cn (Y.L.); btaylor@olivefertility.com (E.L.T.)

**Keywords:** S1P, S1P_1_, CCN2, COX2, PGE2, human granulosa cell

## Abstract

CCN1 and CCN2 are members of the CCN family and play essential roles in the regulation of multiple female reproductive functions, including ovulation. Cyclooxygenase-2 (COX2) is a critical mediator of ovulation and can be induced by sphingosine-1-phosphate (S1P) through the S1P_1/3_-mediated Yes-associated protein (YAP) signaling. However, it is unclear whether CCN1 or CCN2 can mediate S1P-induced upregulation of COX2 expression and increase in prostaglandin E2 (PGE2) production in human granulosa-lutein (hGL) cells. In the present study, we investigated the effects of S1P on the expressions of CCN1 and CCN2 in hGL cells. Additionally, we used a dual inhibition approach (siRNA-mediated silencing and small molecular inhibitors) to investigate the molecular mechanisms of S1P effects. Our results showed that S1P treatment significantly upregulated the expression of CCN1 and CCN2 in a concentration-dependent manner in hGL cells. Additionally, inhibition or silencing of S1P_1_, but not S1P_3_, completely abolished the S1P-induced upregulation of CCN2 expression. Furthermore, we demonstrated that S1P-induced nuclear translocation of YAP and inhibition or silencing of YAP completely abolished the S1P-induced upregulation of CCN1 and CCN2 expression. Notably, silencing of CCN2, but not CCN1, completely reversed the S1P-induced upregulation of COX2 expression and the increase in PGE2 production. Thus, CCN2 mediates the S1P-induced upregulation of COX2 expression through the S1P_1_-mediated signaling pathway in hGL cells. Our findings expand our understanding of the molecular mechanism underlying the S1P-mediated cellular activities in the human ovary.

## 1. Introduction

The CCN family consists of six members, namely CCN1/Cyr61 (cysteine-rich 61), CCN2/CTGF (connective tissue growth factor), CCN3/NOV (nephroblastoma overexpressed), CCN4/WIPSP1, CCN5/WIPSP2, and CCN6/WISP3 [1]. The CCN proteins regulate specific physiological functions of cells in response to various growth factors, cytokines, or extracellular matrices through various signaling cascades [2,3]. Accumulated evidence suggests that CCN1–3 play essential roles in the regulation of various reproductive functions [4], whereas little is known about the functional roles of CCN4–6 in the female reproductive system. For instance, CCN1 is predominantly involved in the processes of angiogenesis in the endometrium [4,5,6] and the corpus luteum [7]. Additionally, CCN1 plays an important role in the female reproductive system during embryogenesis and tumorigenesis [8]. CCN2 is another critical intraovarian regulator; CCN2 cKO mice have disrupted follicular development, increased apoptosis of granulosa cells, decreased ovulation rates, and abnormal steroidogenesis in the ovary [9]. Our previous study showed that CCN2 plays a critical role in the regulation of extracellular matrix (ECM) formation and cell proliferation in human granulosa-lutein (hGL) cells [10,11,12]. Taken together, these results suggest that CCN1 and CCN2 are important intraovarian CCN proteins that modulate follicular development and ovulation.

The Hippo signaling system is a conserved pathway that predominantly controls organ size by regulating multiple cellular activities, including proliferation, adhesion, apoptosis, and stem cell self-renewal [13]. Yes-associated protein (YAP) is the key Hippo signaling effector; Hippo signaling restrains cell proliferation by initiating YAP activities [14]. Once the Hippo signaling pathway is disrupted, a decrease in cytoplasmic YAP phosphorylation leads to an increase in the nuclear level of YAP, which associates with transcriptional enhancer factor TEF-1 (TEAD) to regulate the expression of the downstream target genes [15]. Two CCN proteins, CCN1 and CCN2, have been identified as the target genes of YAP [16]. Recent studies have shown that the Hippo signaling pathway is associated with diverse physiological and pathological conditions in the female reproductive system, including the primordial follicle pool, ovarian germline stem cell proliferation, ovulation, and ovarian aging [17,18,19].

Ovulation is a complex and timely process that involves the coordinated expression of abundant genes in the ovary [20]. Cyclooxygenase-2 (COX2) is one of the ovulation-related genes that produces prostaglandin E2 (PGE2); COX2 and PGE2 are the key mediators of ovulation because COX2-deficient mice exhibit defective ovulation [21]. In rats and monkeys, the administration of indomethacin (a nonselective COX inhibitor) or COX2-specific inhibitors blocked ovulation and the treatment with exogenous PGE2 rescued the inhibitory effect [22,23]. These findings suggest that COX2 and PGE2 are the critical intraovarian factors that trigger ovulation in mammals.

Sphingosine-1-phosphate (S1P) is a bioactive sphingolipid metabolite that can promote the maturation of blood vessels during mammalian development [24]. Outside the endothelial system, S1P is detectable at a high level in human follicular fluid and has been shown to regulate various physiological and pathological processes in the reproductive system [25,26,27]. Various cellular effects of S1P are triggered by binding to G-protein-coupled S1P receptors. At present, five S1P receptors (S1P_1–5_) have been identified that mediate the activation or inhibition of various downstream signaling pathways [24]. S1P_1_ and S1P_3_ are expressed at a high level in human granulosa cells [27,28]. The functional role of the S1P system in human granulosa cells remains largely unknown. Our previous studies have shown that S1P induces expression of COX2 and increases the PGE2 production via the S1P_1/3_-mediated YAP signaling in hGL cells [19]. S1P has been shown to increase the expression of COX2 in human granulosa cells; however, the underlying molecular mechanism of this effect remains to be characterized. In ovarian cancer cells, S1P functions as an upstream stimulator of the Hippo signaling pathway to promote cell proliferation by inducing nuclear accumulation of YAP followed by upregulation of CCN1 and CCN2 [29]. However, it is not known whether CCN1 or CCN2 is involved in the S1P-induced increase in COX2 expression and PGE2 production in human granulosa cells. Based on the findings of the previous studies, we hypothesized that either CCN1 or CCN2 could mediate the S1P-induced canonical Hippo signaling pathway that subsequently regulates the expression of COX2 in hGL cells. Therefore, we sought to investigate the involvement of CCN1 and CCN2 in the Hippo signaling pathway in response to S1P and to examine the underlying molecular mechanisms of the response.

## 2. Materials and Methods

### 2.1. Antibodies and Reagents

Anti-CCN1 (#sc-13100), anti-CCN2 (#sc-14939), anti-Lamin B (#sc-6216), and anti-α-Tublin (#sc-23948) antibodies were obtained from Santa Cruz Biotechnology (Santa Cruz, CA, USA). Polyclonal anti-COX2 (#ab52237) antibody was obtained from Abcam (Cambridge, MA, USA). Anti-YAP (#12395) antibody was obtained from Cell Signaling Technology (Beverly, MA, USA). The horseradish peroxidase (HRP)-conjugated goat anti-mouse and goat anti-rabbit IgG were obtained from Bio-Rad Laboratories (Hercules, CA, USA); the horseradish HRP-conjugated donkey anti-goat IgG was obtained from Santa Cruz Biotechnology. S1P was obtained from Avanti Polar Lipids (Alabaster, AL, USA) and was dissolved in PET solution (5% polyethylene glycol, 2.5% ethanol, and 0.8% Tween-80) [30]. S1P_1_ antagonist W146 and S1P_3_ antagonist CAY10444 were from Cayman Chemical Corp. (Ann Arbor, MI, USA) and were dissolved and diluted according to the manufacturer’s instructions. The nuclear and cytoplasmic extraction reagent (Cat. no. 78835) was obtained from Thermo Fisher Scientific (Grand Island, NY, USA).

### 2.2. Culture of Simian Virus 40 Large T Antigen-Immortalized Human Granulosa Cells (SVOG)

In this study, we used nontumorigenic immortalized human granulosa cells (SVOG cells) previously produced by transfection of the simian virus 40 large T antigen into early luteal phase human granulosa cells obtained from women undergoing in vitro fertilization (IVF) [31]. These cells have been proven to display biological responses to a number of various treatments that are similar to the responses of human granulosa cells under physiological conditions [11,32,33,34,35]. The SVOG cells were seeded and cultured in Dulbecco’s modified Eagle medium (DMEM)/F12 medium (Sigma, Oakville, ON, Canada) supplemented with 10% charcoal/dextran-treated fetal bovine serum (FBS; HyClone Laboratories, Logan, UT, USA), 1× GlutaMAX (Thermo Fisher Scientific/Invitrogen), 100 mg/mL streptomycin sulfate (Thermo Fisher Scientific/Invitrogen), and 100 U/mL penicillin (Thermo Fisher Scientific/Invitrogen). The cells were maintained at humidified atmosphere with 5% CO_2_ at 37 °C. The medium was changed every other day, and the cells were maintained in serum-free medium for 24 h before treatment.

### 2.3. Culture of Primary Human Granulosa-Lutein (hGL) Cells

The hGL cells were obtained from patients undergoing IVF who had provided informed consent after the approval of the University of British Columbia Research Ethics Board. The hGL cells were purified using Ficoll-Paque Plus (Amersham Biosciences, Piscataway, NJ, USA) from follicular aspirates collected during oocyte retrieval from women undergoing IVF as described previously [36]. Individual primary cell cultures were composed of cells from a single patient. Purified hGL cells were seeded in 12-well plates and cultured in DMEM/F12 medium supplemented with 10% charcoal/dextran-treated fetal bovine serum (FBS; HyClone Laboratories), 1× GlutaMAX (Thermo Fisher Scientific/Invitrogen), 100 mg/mL streptomycin sulfate (Thermo Fisher Scientific/Invitrogen), and 100 U/mL penicillin (Thermo Fisher Scientific/Invitrogen) in a humidified atmosphere with 5% CO_2_ at 37 °C. The culture medium was replaced on the next day and changed every other day for up to 5 days.

### 2.4. Small Interfering RNA (siRNA) Transfection

To knock down the endogenous CCN1, CCN2, S1P_1_, S1P_3_, and YAP, the cells were cultured to 50% confluency and transfected with 50 nM ON-TARGET plus SMART pool CCN1 siRNA, 25 nM ON-TARGET plus SMART pool CCN2 siRNA, 75 nM ON-TARGET plus SMART pool S1P_1_ siRNA, 75 nM ON-TARGET plus SMARTpool S1P_3_ siRNA, or 50 nM ON-TARGET plus SMART pool YAP siRNA (Dharmacon, Lafayette, CO, USA) by using Lipofectamine RNAiMAX (Invitrogen, Life Technologies). siControl NON-TARGETING pool siRNA (Dharmacon) was used as a negative control. After 6 h incubation, the culture medium was replaced and the cells were cultured in fresh medium for up to 48 h.

### 2.5. Reverse Transcription Quantitative Real-Time PCR (RT-qPCR)

Total RNA was extracted from the cells with Trizol reagent (Invitrogen) according to the manufacturer’s instructions. RNA (2 µg) was reverse transcribed into the first-strand cDNA with random primers and Moloney murine leukemia virus (M-MLV) reverse transcriptase (Promega, Madison, WI, USA).The qPCR was performed on an Applied Biosystems 7300 real-time PCR system equipped with a 96-well optical reaction plate using a 20 μL RT-qPCR reaction containing 1× SYBR Green PCR master mix (Applied Biosystems, Foster City, CA, USA), 20 ng cDNA, and 250 nM specific primers. The mean values were used for the determination of the mRNA expression levels using the comparative ΔCq (ΔCt) method according to the equation 2^−ΔΔCq^ (2^−ΔΔCt^); GAPDH was used as the reference gene. The RT-qPCR samples were assayed in triplicate to achieve good reproducibility. The specificity of each assay was validated using melting curves analysis. The primer sequences used for qPCR are shown in Table 1. All primers were obtained from Thermo Fisher.

### 2.6. Western Blot Analysis

Cytoplasmic and nuclear protein fractionation was performed using the NE-PER nuclear and cytoplasmic extraction reagents (Thermo) according to the manufacturer’s instructions. Cells were washed with cold PBS and lysed in lysis buffer (Cell Signaling Technology) containing protease inhibitor cocktail (Sigma-Aldrich, Oakville, ON, USA). The extracts were centrifuged at 20,000 *g* for 15 min at 4 °C to remove cellular debris and the protein concentrations were quantified using the DC protein assay (Bio-Rad Laboratories Inc.). Equal amounts (50 μg) of protein were separated by 10% sodium dodecyl sulfate polyacrylamide gel electrophoresis (SDS-PAGE) and then electrophoretically transferred onto the PVDF membranes. After the transfer, the membranes were incubated for 1 h in TBST containing 5% nonfat dried milk at room temperature and overnight at 4 °C with the corresponding primary antibody. After washing in TBST, the membranes were incubated for 1 h at room temperature with the corresponding horseradish peroxidase-conjugated secondary antibody. Immunoreactive bands were detected using an enhanced chemiluminescent substrate or a Super Signal West Femto chemiluminescent substrate (Pierce; Thermo Fisher Scientific) and an X-ray film. Intensity of each band was quantified using ImageJ software.

### 2.7. Prostaglandin E2 Enzyme-Linked Immunosorbent Assay (ELISA)

The culture media were collected and centrifuged at 500 *g* for 5 min at 4 °C to remove cellular debris. The PGE2 levels in the culture media were measured using a PGE2-specific ELISA kit (Cayman Chemical) according to the manufacturer’s protocol. The PGE2 levels were normalized to the protein concentrations of the cell lysate. The PGE2 values were normalized to the control group.

### 2.8. Immunofluorescent Staining

Immunofluorescent staining of SVOG cells was performed as described previously [29]. Briefly, the cells were fixed with 4% paraformaldehyde for 15 min and permeated with 0.1% Triton for 10 min. After blocking in a Dako blocking solution for 1 h, the cells were incubated with an anti-YAP primary antibody (1:100 dilution) overnight at 4 °C. A mouse IgG isotype control was used to detect the primary antibody. After washing with PBS, the cells were incubated with an Alexa Fluor 488-conjugated secondary antibody (Invitrogen, 1:500 dilution) for 1 h in the dark. Samples were mounted using a ProLong Gold antifade reagent with DAPI (Invitrogen) for 5 min. The stained cells were imaged using a Leica SP5II laser scanning confocal microscope; a 405-nm laser was used for the detection of DAPI, and a 488-nm laser was used for the detection of Alexa Fluor 488. The 3D stack images were reconstructed with Olympus cellSens image acquisition and analysis software (version 1.5, Tokyo, Japan).

### 2.9. Statistical Analysis

The results are presented as the mean ± SEM of at least three independent experiments performed with different passages of cells. Statistical analyses were performed by one-way ANOVA and Tukey’s multiple comparison test by using GraphPad Prism Software (San Diego, CA, USA). P-values equal to or <0.05 were considered statistically significant.

## 3. Results

### 3.1. S1P Upregulates the Expression of CCN1 and CCN2 in hGL Cells

To investigate the effects of S1P on the expression of CCN1 and CCN2, we used the immortalized hGL (SVOG) cells as a model. The SVOG cells were treated with a vehicle control (PET) or increasing concentrations (0.1, 0.3, 0.5, or 1 μM) of S1P for 1 h; the results showed that S1P significantly increased the mRNA levels of CCN1 (by 2 times) (Figure 1A) and CCN2 (by 3 times) (Figure 2A) in a concentration-dependent manner. Consistent with the mRNA data, the results obtained from the western blot analysis showed that the lysates of the cells treated with S1P for 2 h had elevated protein levels of CCN1 (by 1.5 times) (Figure 1B) and CCN2 (by 8–10 times) (Figure 2B). The concentrations of S1P in the serum and follicular fluid are approximately 0.9 μM and 0.17 μM, respectively [37]; hence, we treated the cells with 0.3 μM S1P. The results from the time course of the effect showed that 0.3 μM S1P treatment induced an increase in the mRNA levels of CCN1 (by 2 times) (Figure 1C) and CCN2 (by 3 times) (Figure 2C) after 30 min that lasted until 1 h after S1P addition; the levels subsequently decreased thereafter. Results of the western blot analysis showed that S1P induced an increase in the protein levels of CCN1 (by 1.5 times) (Figure 1D) and CCN2 (by 6–8 times) (Figure 2D) 1 h and 2 h after treatment. To further confirm the stimulatory effects of S1P on the expression of CCN1 and CCN2, we used the primary hGL cells that were obtained from patients undergoing IVF procedure. Consistent with the results in the SVOG cells, S1P treatment significantly increased mRNA (Figure 1E and Figure 2E) and protein (Figure 1F and Figure 2F) levels of CCN1 (by 3 times) and CCN2 (by 3–4 times) in primary hGL cells.

### 3.2. The S1P_1_ Receptor Mediates the S1P-Induced Upregulation of CCN1 and CCN2 Expression in hGL Cells

Our previous studies have shown that two S1P receptors, S1P_1_ and S1P_3_, mediate S1P-induced upregulation of COX2 expression in hGL cells [19]. However, it is not known whether S1P_1_ or S1P_3_ mediates the S1P-induced upregulation of CCN1 and CCN2 expression in hGL. To further investigate the underlying mechanisms of S1P-induced regulation of the expression of CCN1 and CCN2, we used two inhibitory approaches, including the utilization of pharmacological inhibitors and specific siRNA-mediated silencing of S1P_1_ and S1P_3_. Two low molecular weight inhibitors, W146 (an antagonist of S1P_1_) and CAY10444 (an antagonist of S1P_3_), were used to block the binding of S1P to S1P_1_ and S1P_3_, respectively. The results showed that pretreatment of the hGL cells (SVOG or primary hGL cells) with 10 μM W146 for 1 h completely reversed the S1P-induced increase in the mRNA and protein levels CCN1 and CCN2 (Figure 3A–C). However, pretreatment of the hGL cells (SVOG or primary hGL cells) with CAY10444 did not alter the S1P-induced increase in the mRNA and protein levels of CCN1 or CCN2 (Figure 3D–F).

Considering the possible off-target effects of pharmacological inhibitors, we used specific siRNA-mediated silencing of endogenous expression of the targets to determine which S1P receptor (S1P_1_ or S1P_3_) mediates the S1P-induced upregulation of CCN1 and CCN2 expression in SVOG cells. As shown in Figure 4A,B, silencing efficiency studies demonstrated that transfection with specific siRNAs targeting endogenous S1P_1_ or S1P_3_ significantly decreased the mRNA levels of S1P_1_ (Figure 4A) and S1P_3_ (Figure 4B). Notably, the silencing of S1P_1_ partially abolished the S1P-induced increase in the mRNA (Figure 4A) and protein (Figure 4C) levels of CCN1 and CCN2, whereas the silencing of S1P_3_ did not induce these effects (Figure 4B,D). These results indicate that S1P_1_ is the predominant S1P receptor required for an increase in the expression of CCN1 and CCN2 in response to S1P in hGL cells.

### 3.3. S1P Induces Nuclear Translocation of YAP in SVOG Cells

Our previous studies have demonstrated that S1P treatment for 30 min induced rapid de-phosphorylation at YAP S127 in SVOG cells [19]; however, there is no evidence that S1P can induce nuclear translocation of YAP in hGL cells. To verify the effect of S1P on the intracellular activity of YAP, we determined the subcellular distribution of YAP after S1P treatment using immunofluorescence staining and analyzed the YAP protein contents in the nucleus using a nuclear extraction reagent. In the control group, the majority of YAP staining (red) and DAPI staining (green) were not colocalized; however, S1P treatment significantly increased intranuclear YAP staining (red) that was colocalized with DAPI (green) under the presented 3D image (Figure 5A), indicating that S1P treatment induced translocation of YAP into the nucleus. Additionally, the results of the western blot analysis showed that S1P significantly increased the nuclear retention of YAP (using nuclear lamin B as the loading control) in SVOG cells (Figure 5B).

### 3.4. YAP Mediates the S1P-Induced Upregulation of CCN1, CCN2, and COX2 Expression

To investigate the involvement of YAP in the S1P-induced upregulation of CCN1 and CCN2 expression, we used a small molecular inhibitor (Verteporfin, VP) that blocks the formation of the YAP and TEAD complex. The result showed that the S1P-induced increase in the mRNA (Figure 6A) and protein (Figure 6B) levels of CCN1, CCN2, and COX2 was completely abolished by pretreatment of the SVOG cells with 10 μM VP. We then used specific siRNA targeting YAP to confirm the functional role of YAP in hGL cells. As shown in Figure 6C,D, the silencing of YAP completely abolished the S1P-induced increase in mRNA and protein levels of CCN1 and CCN2 in SVOG cells. These results indicate that YAP mediates the S1P-induced upregulation of CCN1 and CCN2 expression in hGL cells.

### 3.5. CCN2, But Not CCN1, Mediates the S1P-Induced Upregulation of COX2 and Increase in PGE2 Production in SVOG Cells

Our previous studies have shown that S1P can upregulate COX2 expression and, in turn, increase PGE2 production in hGL cells [19]. To determine whether CCN1 and CCN2 are required for the S1P-induced upregulation of COX2 expression in hGL cells, we used siRNA-mediated silencing to target endogenous CCN1 and CCN2 in SVOG cells. The CCN1 and CCN2 silencing efficiencies were examined using RT-qPCR (Figure 7A,B). Notably, silencing of CCN2 did not influence the basal levels of COX2 expression; however, silencing of CCN2 completely reversed the S1P-induced increase in mRNA (Figure 7B) and protein (Figure 7D) levels of COX2 in SVOG cells. However, silencing of CCN1 did not influence the S1P-induced increases in the mRNA (Figure 7A) and protein (Figure 7C) COX2 expression levels. Similarly, silencing of CCN2 completely abolished the S1P-induced increase in the mRNA and protein levels of COX2 in primary hGL cells (Figure 7E). PGE2 is the major COX product and has been well characterized as a critical regulator of various reproductive functions in females [38]. To further investigate whether S1P-induced upregulation of CCN2 expression contributes to the increase in PGE2 production, we examined the concentration of PGE2 in conditioned medium (spent media harvested from cultured cells) after specific treatments. The ELISA results showed that S1P treatment significantly increased PGE2 levels in the conditioned medium starting at 3 h incubation, and this increase persisted until 12 h after the treatment (Figure 7F). Notably, silencing of CCN2 completely abolished the S1P-induced increase in PGE2 production in conditioned medium in both SVOG and primary hGL cells (Figure 7F,G). These results indicate that CCN2, but not CCN1, is required for the S1P-induced upregulation of COX2 expression and the increase in PGE2 production in hGL cells.

## 4. Discussion

Ovulatory disorders are one of the leading causes of female infertility and are diagnosed in approximately 25% of women who experience fertility problems [39]. To enhance our understanding of the pathogenesis and diversity of ovulatory disorders, a comprehensive molecular analysis of the ovulation-related signaling network is urgently required for the development of individual therapeutic strategies. Considering the importance of COX-2 for PGE2 formation and subsequent ovulation, a number of studies have investigated the regulation of COX2 in ovarian follicles [21,40,41,42]. In particular, our previous studies have demonstrated the functional role of S1P in the regulation of COX2 in hGL cells [19]. However, the detailed molecular mechanisms by which S1P upregulates the expression of COX2 have not been elucidated. In this study, using primary and immortalized hGL cells as models, we demonstrated that treatment with exogenous S1P upregulated the expressions of CCN1 and CCN2 at both the transcriptional and translational levels.

The two S1P receptors, S1P_1_ and S1P_3_, are expressed at a high level in the human ovary [27,28]; hence, we determined which S1P receptor mediates the S1P-induced effects in hGL cells. Using a dual inhibition approach, we found that inhibition of S1P_1_ (using an S1P_1_ inhibitor W146) or silencing of S1P_1_ (using siRNAs targeting S1P_1_) attenuated the S1P-induced stimulation of the expression of CCN1 and CCN2. However, inhibition of S1P_3_ (using an S1P_3_ inhibitor CAY10444) or silencing of S1P_3_ (using siRNAs targeting S1P_3_) had no effect. These results indicate that S1P_1_, but not S1P_3_, is the functional receptor that mediates the S1P-induced expression of CCN proteins in hGL cells. The results of the previous studies are not consistent with the data of the present study and showed that S1P enhances the proliferation of hepatocellular carcinoma cells by upregulating the expression of CCN2 via the S1P_2_-mediated intracellular signaling pathway [43]. In this regard, S1P exerts its effects by using different S1P receptors (S1P_1_ or S1P_2_) in different tissues or cells. Therefore, the specific form of the S1P receptor that mediates the S1P-induced effects may be cell-type dependent.

In certain mammalian cells, S1P is the upstream regulator of the Hippo pathway that induces a decrease in phosphorylated YAP and subsequent nuclear translocation of YAP that eventually interacts with other nuclear factors to regulate the target genes [29]. Our previous studies have shown that S1P treatment significantly decreased the cellular level of phosphorylated YAP in hGL cells [19]. In the present study, using the immunofluorescent assay, we showed that S1P treatment significantly increased intranuclear YAP staining, indicating that S1P treatment induced translocation of YAP into the nucleus. Additionally, the expression of YAP in cell fractions after the S1P treatment clearly showed that S1P significantly increases the nuclear level of YAP in hGL cells. The results of the present study are consistent with the data of our previous studies and showed that S1P can induce nuclear transfer of YAP in ovarian cancer cells [29]. Using a similar inhibition approach, we showed that silencing of YAP with siRNAs targeted YAP or inhibited the association of YAP with the downstream TEAD factors; the inhibitor VP completely abolished the S1P-induced upregulation of CCN1 and CCN2 expression in hGL cells. These results indicate that YAP is the intracellular upstream effector that mediates the S1P-induced effects on the expression of the CCN proteins.

In the present study, we demonstrated that S1P promptly upregulates the expression of CCN1 and CCN2 as early as 30 min after the initiation of the S1P treatment and that the stimulatory effects gradually subside after reaching a peak level at 2 h after treatment. The stimulatory effects of S1P on an increase in the expression of CCN1 and CCN2 are substantially faster than those on the expression of COX2, indicating that CCN proteins are the upstream effectors of COX2 upregulation. Previous studies have shown that CCN1 and CCN2 are simultaneously upregulated in response to various stimulants, which, in turn, modulate multiple functions in certain mammalian cells [44,45,46]. Considering the critical roles of CCN1 and CCN2 in the regulation of follicular development and ovulation, we sought to investigate the involvement of CCN proteins in the S1P-induced upregulation of COX2. Using the siRNA-mediated silencing approach, we demonstrated that silencing of CCN2, but not CCN1, completely abolished the stimulatory effect of S1P on the expression of COX2 in primary and immortalized hGL cells. Additionally, functional studies confirmed that silencing of CCN2 completely abolished the increase in PGE2 accumulation in the conditioned medium in response to S1P. These results indicate that CCN2 is the unique CCN protein that mediates the S1P-induced upregulation of COX2 and the increase in PGE2 production in hGL cells. Similarly, CCN2 mediates the S1P-induced angiogenic effect in human dermal microvascular endothelial cells [47]. However, inconsistent with these results, our previous studies showed that both CCN1 and CCN2 mediate the cellular action downstream of S1P in ovarian cancer cells [29]. Despite structural similarities, CCN1 and CCN2 may have different regulatory functions due to binding to different cofactors or responding to diverse microenvironments [3]. The limitation of this study is that we did not demonstrate the detailed molecular mechanism by which CCN2 mediates the S1P-induced upregulation of COX2 expression in hGL cells. After the translation of the CCN superfamily members, the mature proteins are secreted and bind to the integrin receptors to trigger the downstream signaling pathway and subsequent gene regulation [48]. Future studies should be focused on the connection and interaction of the CCN2 protein and the related membrane receptors in the developing follicles using an in vivo animal model; these studies will be of great interest.

In conclusion, our study showed that S1P induces significant upregulation of CCN1 and CCN2 in hGL cells. Additionally, this biological function is regulated via the S1P_1_-mediated nuclear translocation of the intracellular effector YAP. Furthermore, the results obtained by using the inhibition approach showed that CCN2, but not CCN1, mediates the S1P-induced upregulation of COX2 and the increase in PGE2 production that occurs in hGL cells (Figure 8). These findings provide valuable information on the molecular mechanisms of the S1P-induced effects in hGL cells, which provides a better understanding of the roles of the Hippo signaling in human follicular function.

## Figures and Tables

**Figure 1 cells-08-01445-f001:**
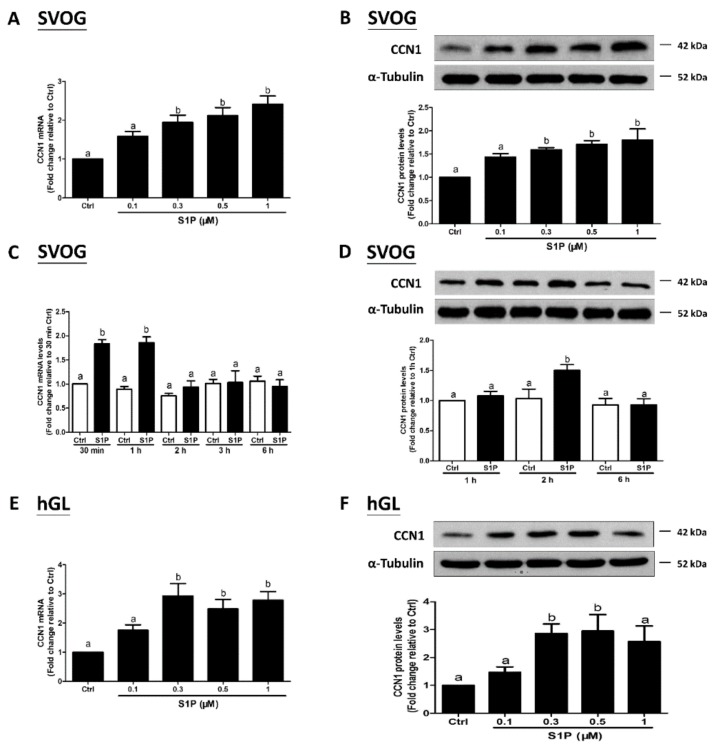
Sphingosine-1-phosphate (S1P) induces the expression of CCN1 in human granulosa-lutein cells. Immortalized human granulosa-lutein (SVOG) cells were treated with a vehicle control (PET solution containing 5% polyethylene glycol, 2.5% ethanol, and 0.8% Tween-80) or increasing concentrations (0.1, 0.3, 0.5, or 1 μM) of S1P for 1 h (**A**) or 2 h (**B**). The mRNA and protein levels of CCN1 were examined using RT-qPCR (**A**) and western blot analysis (**B**), respectively. SVOG cells were treated with a vehicle control or 0.3 μM S1P for various periods of time (30 min, 1 h, 2 h, 3 h, or 6 h); the mRNA and protein levels of CCN1 were examined using RT-qPCR (**C**) and western blot analysis (**D**), respectively. primary human granulosa-lutein (hGL) cells were treated with a vehicle control or increasing concentrations (0.1, 0.3, 0.5, and 1 μM) of S1P for 1 h (**E**) or 2 h (**F**). The mRNA and protein levels of CCN1 were examined using RT-qPCR (**E**) and western blot analysis (**F**), respectively. The results are presented as the mean ± SEM of at least three independent experiments. If a pair of values is significantly different (*p* < 0.05), the values have different subscript letters (a vs. b or b vs. c) assigned to them. Ctrl, control; S1P, sphingosine-1-phosphate.

**Figure 2 cells-08-01445-f002:**
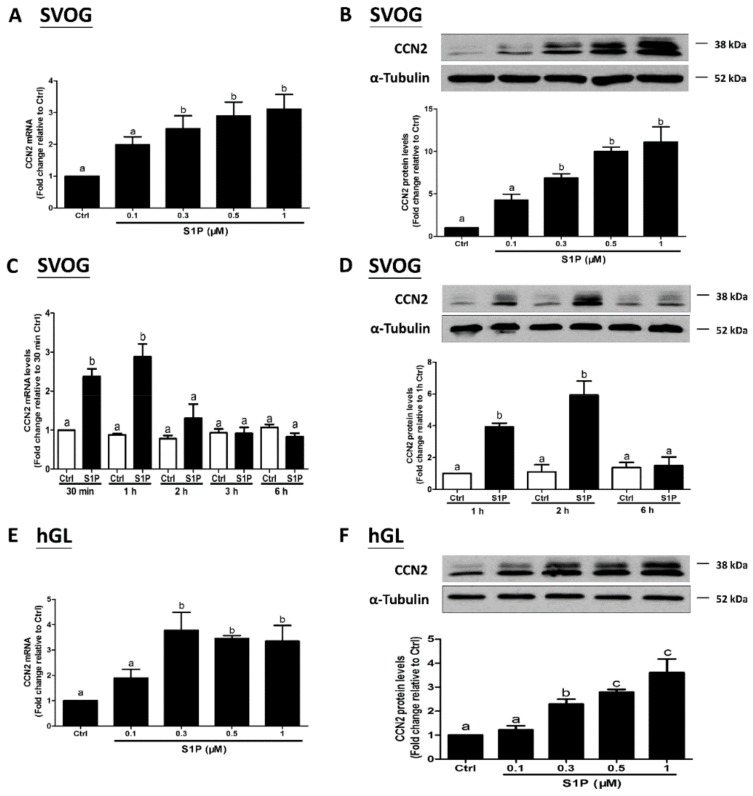
S1P induces the expression of CCN2 in human granulosa-lutein cells. SVOG cells were treated with a vehicle control or increasing concentrations (0.1, 0.3, 0.5, and 1 μM) of S1P for 1 h (**A**) or 2 h (**B**). The mRNA (**A**) and protein (**B**) levels of CCN2 were examined using RT-qPCR (**A**) and western blot analysis (**B**), respectively. SVOG cells were treated with a vehicle control or 0.3 μM S1P for various periods of time (30 min, 1 h, 2 h, 3 h, or 6 h); the mRNA (**C**) and protein (**D**) levels of CCN2 were examined using RT-qPCR (**C**) and western blot analysis (**D**), respectively. Primary human granulosa-lutein (hGL) cells were treated with a vehicle control or increasing concentrations (0.1, 0.3, 0.5, or 1 μM) of S1P for 1 h (**E**) or 2 h (**F**). The mRNA (**E**) and protein (**F**) levels of CCN2 were examined using RT-qPCR (**E**) and western blot analysis (**F**), respectively. The results are presented as the mean ± SEM of at least three independent experiments. If a pair of values is significantly different (*p* < 0.05), the values have different subscript letters (a vs. b or b vs. c) assigned to them.

**Figure 3 cells-08-01445-f003:**
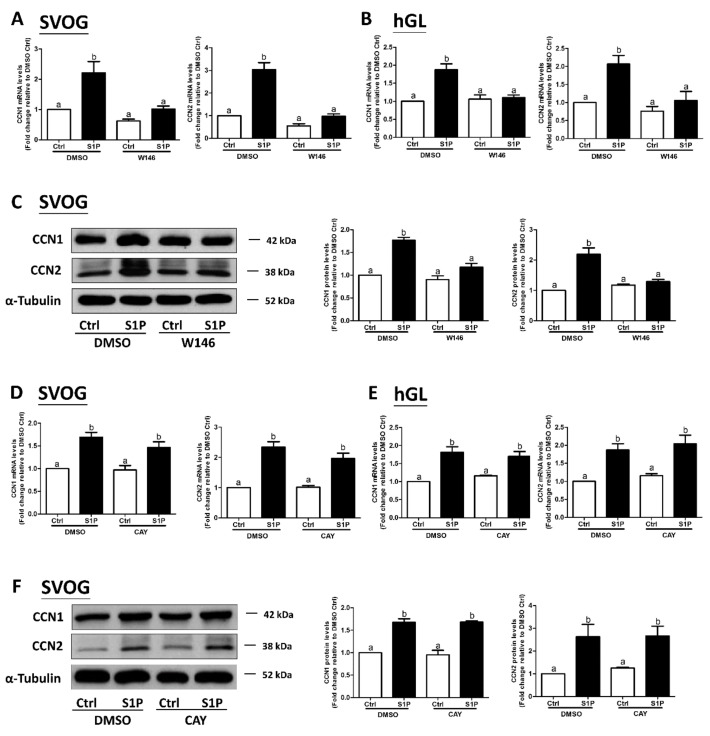
The S1P_1_ receptor antagonist W146 abolishes the S1P-induced upregulation of CCN1 and CCN2 in human granulosa-lutein cells. SVOG cells (**A**) and primary human granulosa-lutein (hGL) cells (**B**) were pretreated with a vehicle control (DMSO) or 10 μM W146 (an antagonist of S1P_1_) for 1 h and then treated with a vehicle control or 0.3 μM S1P for 1 h. The mRNA levels of CCN1 (**A**) and CCN2 (**B**) were examined using RT-qPCR. (**C**) SVOG cells were pretreated with a vehicle control or 10 μM W146 for 1 h and then treated with a vehicle control or 0.3 μM S1P for 2 h. The protein levels of CCN1 and CCN2 were examined using western blot analysis. SVOG cells (**D**) and primary human granulosa-lutein cells (**E**) were pretreated with a vehicle control or 10 μM CAY10444 (an antagonist of S1P_3_) for 1 h and then treated with a vehicle control or 0.3 μM S1P for 1 h. The mRNA levels of CCN1 and CCN2 were examined using RT-qPCR. (**F**) SVOG cells were pretreated with a vehicle control or 10 μM CAY10444 for 1 h and then treated with a vehicle control or 0.3 μM S1P for 2 h. The protein levels of CCN1 and CCN2 were examined using western blot analysis. The results are presented as the mean ± SEM of at least three independent experiments. If a pair of values is significantly different (*p* < 0.05), the values have different subscript letters (a vs. b or b vs. c) assigned to them.

**Figure 4 cells-08-01445-f004:**
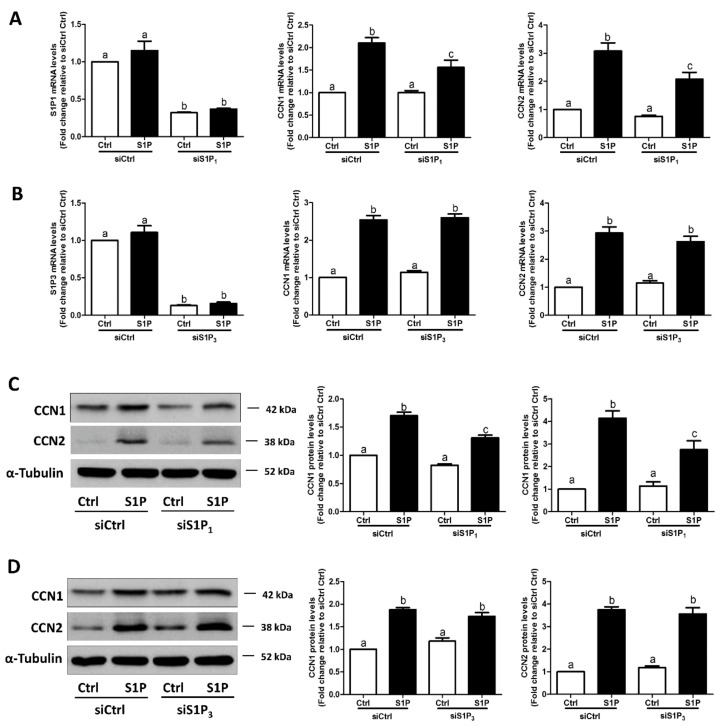
S1P_1_ mediates the S1P-induced upregulation of CCN1 and CCN2 expression in SVOG cells. (**A**) SVOG cells were transfected with 75 nM control siRNA (siCtrl) or 75 nM S1P_1_ siRNA (siS1P_1_) for 48 h, followed by treatment with a vehicle control or 0.3 μM S1P for 1 h. The mRNA levels of S1P_1_, CCN1, and CCN2 were examined using RT-qPCR. (**B**) SVOG cells were transfected with 75 nM siCtrl or 75 nM S1P_3_ siRNA (siS1P_3_) for 48 h, followed by treatment with a vehicle control or 0.3 μM S1P for 1 h. The mRNA levels of S1P_3_, CCN1, and CCN2 were examined using RT-qPCR. (**C**) SVOG cells were transfected with 75 nM siCtrl or 75 nM siS1P_1_ for 48 h, followed by treatment with a vehicle control or 0.3 μM S1P for 2 h. The protein levels of CCN1 and CCN2 were examined using western blot analysis. (**D**) SVOG cells were transfected with 75 nM siCtrl or 75 nM siS1P_3_ for 48 h, followed by treatment with a vehicle control or 0.3 μM S1P for 2 h. The protein levels of CCN1 and CCN2 were examined using western blot analysis. The results are presented as the mean ± SEM of at least three independent experiments. If a pair of values is significantly different (*p* < 0.05), the values have different subscript letters (a vs. b or b vs. c) assigned to them.

**Figure 5 cells-08-01445-f005:**
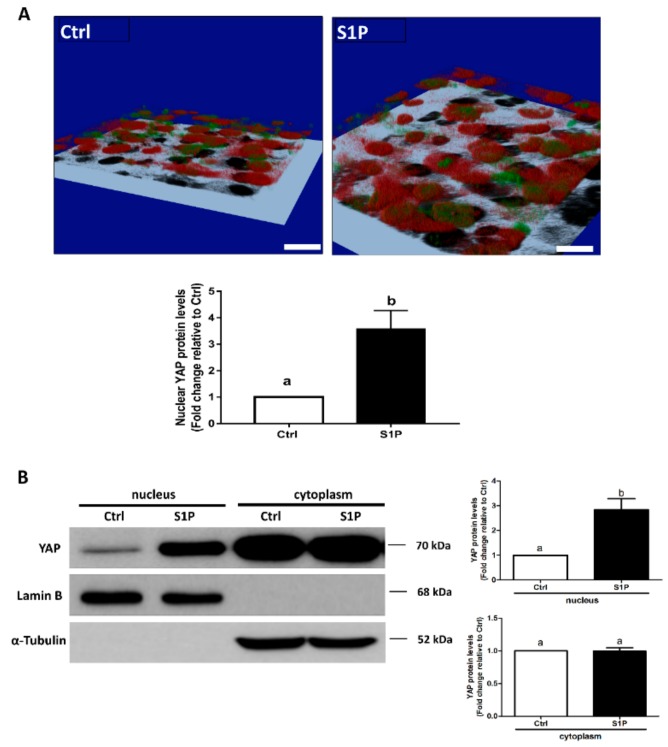
S1P induces nuclear translocation of Yes-associated protein (YAP) in SVOG cells. (**A**) SVOG cells were treated with a vehicle control or 0.3 μM S1P for 30 min; the subcellular localization of YAP was determined using immunofluorescence staining for endogenous YAP (red) along with DAPI staining to show nuclei (green); the 3D images were acquired with a confocal microscope. The yellow pixels of the colocalized region indicate the intranuclear localization of YAP. Black speckles indicate the shadows of SVOG cells. Scale bars represent 50 μm. (**B**) SVOG cells were treated with a vehicle control or 0.3 μM S1P for 1 h. The protein levels of YAP in the cytoplasmic and nuclear fractions were examined using western blot analysis. Lamin B and α-Tubulin were used as the control proteins for the cytoplasmic and nuclear fractions, respectively. The results are presented as the mean ± SEM of at least three independent experiments. If a pair of values is significantly different (*p* < 0.05), the values have different subscript letters (a vs. b or b vs. c) assigned to them.

**Figure 6 cells-08-01445-f006:**
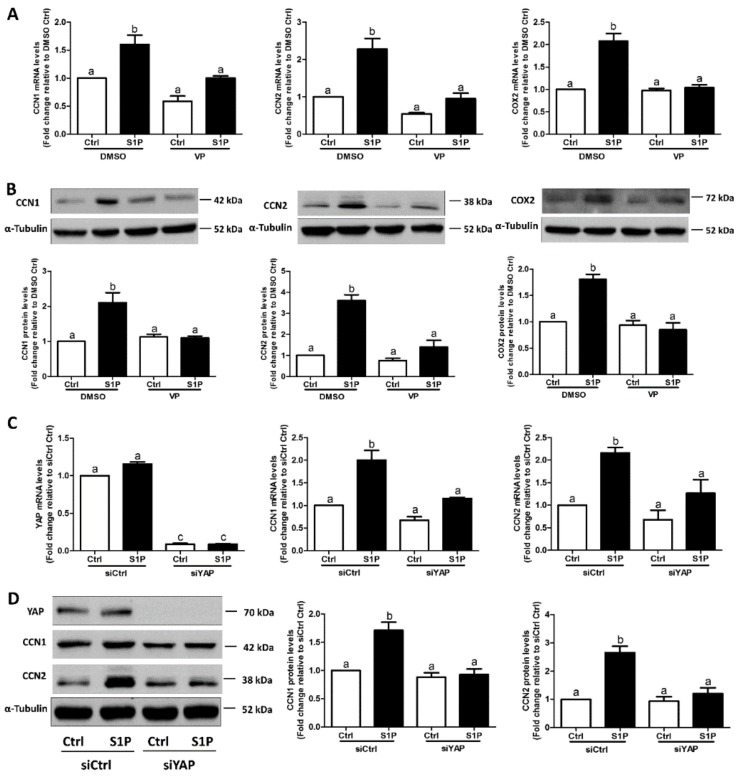
YAP mediates the S1P-induced upregulation of CCN1, CCN2, and COX2 expression in SVOG cells. SVOG cells were pretreated with a vehicle control (DMSO) or 10 μM verteporfin (VP, a YAP-specific inhibitor) for 1 h and then treated with a vehicle control or 0.3 μM S1P for 1 h (**A**) or 2 h (**B**). The mRNA (**A**) and protein (**B**) levels of CCN1, CCN2, and COX2 were examined using RT-qPCR (**A**) and western blot analysis (**B**), respectively. SVOG cells were transfected with 50 nM siCtrl or 50 nM YAP siRNA (siYAP) for 48 h. followed by treatment with a vehicle control or 0.3 μM S1P for 1 h (**C**) or 2 h (**D**). The mRNA (**C**) and protein (**D**) levels of YAP, CCN1 and CCN2 were examined using RT-qPCR (**C**) and western blot analysis (**D**), respectively. The results are presented as the mean ± SEM of at least three independent experiments. Values marked by different letters were significantly different (*p* < 0.05).

**Figure 7 cells-08-01445-f007:**
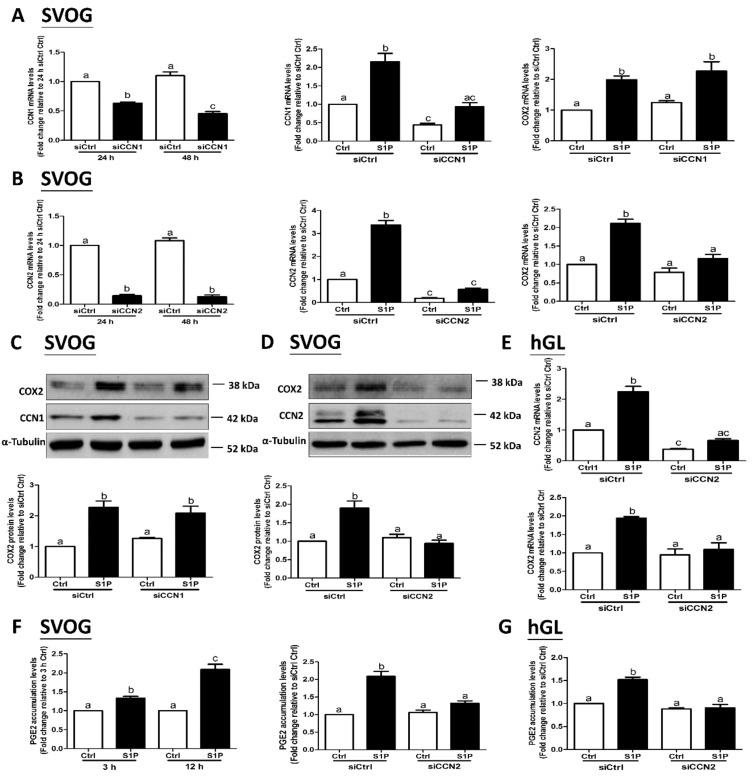
CCN2, but not CCN1, mediates the S1P-induced upregulation of COX2 and the increase in PGE2 production in human granulosa-lutein cells. (**A**) SVOG cells were transfected with 50 nM siCtrl or 50 nM CCN1 siRNA (siCCN1) for 48 h, followed by treatment with a vehicle control or 0.3 μM S1P for 1 h. The mRNA levels of CCN1 and COX2 were examined using RT-qPCR. (**B**) SVOG cells were transfected with 25 nM siCtrl or 25 nM CCN2 siRNA (siCCN2) for 48 h, followed by treatment with a vehicle control or 0.3 μM S1P for 1 h. The mRNA levels of CCN2 and COX2 were examined using RT-qPCR. SVOG cells were transfected with 25 nM or 50 nM control siRNA (siCtrl), 50 nM siCCN1 (**C**), or 25 nM siCCN2 (**D**) for 48 h, followed by treatment with a vehicle control or 0.3 μM S1P for 2 h (for CCN1 and CCN2) and 12 h (for COX2). The protein levels of CCN1 (**C**), CCN2 (**D**) and COX2 (**C** and **D**) were examined using western blot analysis. (**E**) primary human granulosa-lutein (hGL) cells were transfected with 25 nM siCtrl or 25 nM siCCN2 for 48 h, followed by treatment with a vehicle control or 0.3 μM S1P for 1 h (for CCN2) or 12 h (for COX2). The mRNA levels of CCN2 and COX2 were examined using RT-qPCR. (**F**) SVOG cells were transfected with 25 nM siCtrl or siCCN2 for 48 h followed by the treatment with a vehicle control or 0.3 μM S1P for 12 h. The accumulation of PGE2 in the culture medium was measured using ELISA. (**G**) hGL cells were transfected with 25 nM siCtrl or siCCN2 for 48 h, followed by treatment with a vehicle control or 0.3 μM S1P for 12 h. The accumulation of PGE2 in the culture medium was measured using ELISA. The results are presented as the mean ± SEM of at least three independent experiments. If a pair of values is significantly different (*p* < 0.05), the values have different subscript letters (a vs. b or b vs. c) assigned to them. COX2, cyclooxygenase-2; PGE2, prostaglandin E2.

**Figure 8 cells-08-01445-f008:**
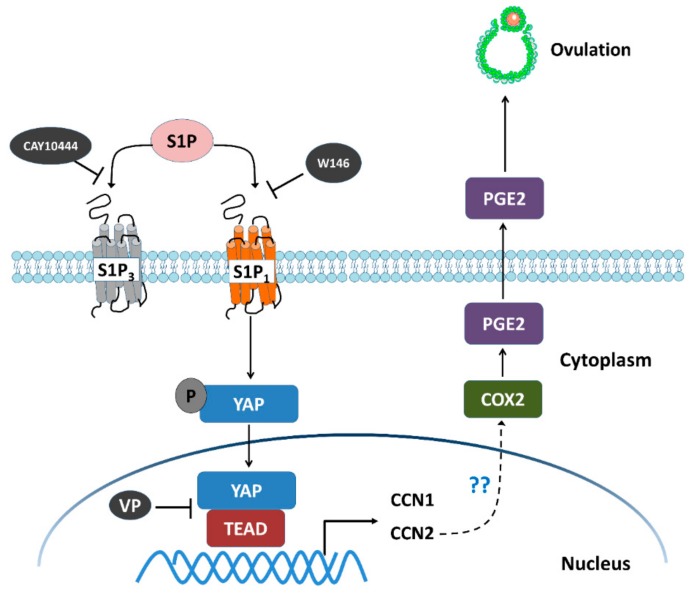
Schematic diagram representing the molecular mechanism underlying the S1P-induced upregulation of COX2 expression and the increase in PGE2 production in human granulosa-lutein cells. S1P binds to S1P receptors S1P_1_ or S1P_3_ in human granulosa cells. Combination of S1P and S1P_1_ induces nuclear translocation of YAP, which, in turn, upregulates the expression of CCN1 and CCN2. The increased level of CCN2, but not CCN1, upregulates the expression of COX2 and increases the production of PGE2, which eventually induces ovulation. W146 (an antagonist of S1P_1_) or verteporfin (VP, an antagonist of YAP), but not CAY10444 (an antagonist of S1P_3_), significantly inhibit the S1P-induced upregulation of COX2 expression and the increase in PGE2 production.

**Table 1 cells-08-01445-t001:** Sequences of primer for RT-qPCR used in this study.

Name	Forward (5’–3’)	Reverse (5’–3’)
CCN1	AGCCTCGCATCCTATACAACC	TTCTTTCACAAGGCGGCACTC
CCN2	GCGTGTGCACCGCCAAAGAT	CAGGGCTGGGCAGACGAACG
COX2	CCCTTGGGTGTCAAAGGTAA	GCCCTCGCTTATGATCTGTC
S1P_1_	TGCGGGAAGGGAGTATGTTT	CGATGGCGAGGAGACTGAAC
S1P_3_	TGCAGCTTCATCGTCTTGGAG	GCCAATGAAAAAGTACATGCGG
GAPDH	GAGTCAACGGATTTGGTCGT	GACAAGCTTCCCGTTCTCAG

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
