# Peer review of "CCN2 Mediates S1P-Induced Upregulation of COX2 Expression in Human Granulosa-Lutein Cells"

_cells, 2019, doi:10.3390/cells8111445_

Round 1

Reviewer 1 Report

The manuscript is well elaborated. The results are new and the topic is interesting. The findings represent a promising impact on further implementation in the therapy. However, I have some comments:

application of siRNA results to the silencing of the selected gene and not a knockdown, this should be corrected. representation of colocalized YAP and DAPI is confused. Usually, if green is colocalized with red the results represent yellow-colored pixels of colocalized region. Moreover, you can quantify the level of colocalization. Did you perform it?

I suggest accepting the manuscript after minor revision.

Author Response

Response to Reviewer 1 Comments

The manuscript is well elaborated. The results are new and the topic is interesting. The findings represent a promising impact on further implementation in the therapy. However, I have some comments:

application of siRNA results to the silencing of the selected gene and not a knockdown, this should be corrected. representation of colocalized YAP and DAPI is confused. Usually, if green is colocalized with red the results represent yellow-colored pixels of colocalized region. Moreover, you can quantify the level of colocalization. Did you perform it?

I suggest accepting the manuscript after minor revision.

Response:

We thank the reviewer’s positive comments and insightful suggestion. As per the reviewer’s suggestion, we have revised the manuscript by replacing the “knockdown” by “silencing”. Additionally, we have revised the figure legend of Figure 5A and performed the quantification of the fluorescent staining (please see the revised Figure 5A).

Reviewer 2 Report

Manuscript ID: cells-621240

Title: CCN2 mediates S1P-induced upregulation of COX2 2 expression in human granulosa-lutein cells

The submitted manuscript presents an article about the mediation of the S1P upregulation of COX2 expression by CCN2 in SVOG cells and hGL cells. The presented work establish this mediation through the use of specific inhibitors and siRNA approach.

The article fits well into the area of the Journal. The article is overall very well written and easy to follow by the reader.

Style

P3, L96                  A space mark should be inserted between “S1P” and “obtained”.

P6, Fig.1/L218    It is no clear what is the meaning of letters. For example, Fig. 1A: “a” significance to the control and “b” significance to the 0.1 µM treatment? The explanation in the legend must be rewritten to be clearer.

                                This remark applies to figures 1 to 7.

Content:

P4, L138               2 mg of RNA seems too much for a RT step. I think it must be 2 µg. Needs to be corrected.

P5, Chap. 3.1      In order to be more rigorous, numerical values of the increases of mRNAs and proteins should be added in the text. Example: for CCN1 mRNA increases by 2 times and the protein by 1.5 time in SVOG, in hGL the increase is higher with 3 times for each experiment. I think that this latter point should noticed in the text.

P5, L200/201      The phrase is not clear enough here. It suggests that the increase of CCN2 protein occurs only at 2h while it occurs significantly at 1h and 2h (Fig.2D). Please correct the phrase.

P6, Fig.1B/D and P7, Fig.2B/D

                                The molecular weights of CCN1/CCN2 and alpha-tubulin are not stated in western blot pictures. Please add.

                                The entire western blot pictures with markers should be provided in supplementary data in order to validate the size of the bands.

P7, Fig.2B/D       The western blot for CCN2 shows two bands. Which one corresponds to CCN2? Both? Which band was used for densitometry?

P10, Fig.4            Do you have an explanation for the quasi absence of western blot signal for CCN2 protein in controls (fig.4C) while it is visible in other experiments?

                                Is there a difference in deltaCt for controls in QPCR experiments between CCN1 and CCN2?

P11, Fig5A           This part is really difficult to understand and consequently to agree with authors interpretations. The two images are not presented with the same orientation (viewing angle) and there is no graphical supports (arrows to point features to look at). In addition, in classical co-localization confocal microscopic studies, co-localization is visible by a yellow color (mix of green and red colors) that is not presented here. I think that this part should be presented differently (3D?/2D?) or not presented if co-localization is not clearly established and the text (P11, L291/293 and P15,L401/403) must be adapted consequently.

P13, L344/345/348 and P15, L425

                                The authors mention “conditioned medium”. To be clear can the authors explain the meaning of “conditioned” in the context?

P15, L415/416    The authors deals with the rapidity of the stimulatory effect of S1P on expression of CCN1 and CCN2 in comparison with COX2 expression. For the latter, the time is not mentioned: to be accurate the time (48h) must be given here.

P16, Fig.8 and P15, L432/434

                                As stated in the discussion about the absence of a detailed mechanism of CCN2 mediation of the COX2 expression, Figure 8 should be slightly modify: the arrow linking CCN2 to COX2 should be dotted (and not in plain line) and marked with a question mark. With the actual scheme it seems that CCN2 acts directly on COX2 after exit from the nucleus while, as explain in P15, L435, it should be secreted and acts indirectly on COX2 expresion.

Author Response

Response to Reviewer 2 Comments

The submitted manuscript presents an article about the mediation of the S1P upregulation of COX2 expression by CCN2 in SVOG cells and hGL cells. The presented work establish this mediation through the use of specific inhibitors and siRNA approach.

The article fits well into the area of the Journal. The article is overall very well written and easy to follow by the reader.

Response:

We thank the reviewer’s positive comments. 

Style

P3, L96                  A space mark should be inserted between “S1P” and “obtained”.

Response:

We have revised the manuscript accordingly (please see line 111).

P6, Fig.1/L218    It is no clear what is the meaning of letters. For example, Fig. 1A: “a” significance to the control and “b” significance to the 0.1 µM treatment? The explanation in the legend must be rewritten to be clearer.

                                This remark applies to figures 1 to 7.

Response:

We have revised the manuscript by clarifying the meaning of letters (please see the revised figure legends).

“If a pair of values is significantly different (p<0.05), the values have different subscript letters (a vs. b or b vs. c) assigned to them.”

Content:

P4, L138               2 mg of RNA seems too much for a RT step. I think it must be 2 µg. Needs to be corrected.

Response:

We apologize for the mistake and have revised the manuscript accordingly (please see line 153).

P5, Chap. 3.1      In order to be more rigorous, numerical values of the increases of mRNAs and proteins should be added in the text. Example: for CCN1 mRNA increases by 2 times and the protein by 1.5 time in SVOG, in hGL the increase is higher with 3 times for each experiment. I think that this latter point should noticed in the text.

Response:

As per the reviewer’s suggestion, we have revised the manuscript accordingly (please see lines 206-220).

P5, L200/201      The phrase is not clear enough here. It suggests that the increase of CCN2 protein occurs only at 2h while it occurs significantly at 1h and 2h (Fig.2D). Please correct the phrase.

Response:

As per the reviewer’s suggestion, we have revised the manuscript accordingly (please see lines 216).

P6, Fig.1B/D and P7, Fig.2B/D

                                The molecular weights of CCN1/CCN2 and alpha-tubulin are not stated in western blot pictures. Please add.

                                The entire western blot pictures with markers should be provided in supplementary data in order to validate the size of the bands.

Response:

As per the reviewer’s suggestion, we have revised the western blot pictures by adding the indicated molecular weights of the corresponding antibody targets (please see the revised figures).

P7, Fig.2B/D       The western blot for CCN2 shows two bands. Which one corresponds to CCN2? Both? Which band was used for densitometry?

Response:

Two bands of CCN2 shown in the western blot analysis indicates the 36 to 38 kDa form of CCN2. Glycosylation studies using Hs578T cells (human breast cancer cells) have demonstrated that CCN2 is composed of a 30 kDa protein core containing 2 to 8 kDa of N-linked sugars (Yang et al. J Clin Endocrinol Metab, 1998). Therefore, the 36 to 38 kDa form of CCN2 represents N-glycosylation variants of the intact CTGF peptide (Zarrinkalam et al. Kidney International, 2003). Therefore, both bands were used for densitometry in this study.

P10, Fig.4            Do you have an explanation for the quasi absence of western blot signal for CCN2 protein in controls (fig.4C) while it is visible in other experiments?

Response:

We agree with the reviewer’s point that CCN2 protein should display more than one band on the western blotting images. Indeed, we can visualize one or two bands on our western blotting results dependent on the exposure time. As the following figure (original Figure 4C), we can visualize two bands (red arrowhead) of the CCN2 protein by amplifying the contrast.

CCN2 protein

                                Is there a difference in deltaCt for controls in QPCR experiments between CCN1 and CCN2?

Response:

In this study, we used relative quantification or comparative quantification to measure the relative change in mRNA expression levels. It determines the changes in steady state mRNA levels of a gene across multiple samples and expresses it relative to the levels of another RNA. To determine the level of expression, the differences (∆) between the threshold cycle (Ct) are measured. Thus, the mentioned methods can be summarized as the ∆CP methods (Morse et al., 2005; Livak and Schmittgen, 2001). The average Ct was calculated for CCN1 and GAPDH (or for CCN2 and GAPDH), and the ∆Ct (CtCCN1 – CtGAPDH or CtCCN2 – CtGAPDH) was determined in each sample.

P11, Fig5A           This part is really difficult to understand and consequently to agree with authors interpretations. The two images are not presented with the same orientation (viewing angle) and there is no graphical supports (arrows to point features to look at). In addition, in classical co-localization confocal microscopic studies, co-localization is visible by a yellow color (mix of green and red colors) that is not presented here. I think that this part should be presented differently (3D?/2D?) or not presented if co-localization is not clearly established and the text (P11, L291/293 and P15,L401/403) must be adapted consequently.

Response:

Indeed, it is difficult to interpret the immunochemistry 2D-image by merely determining the color change. Therefore, we presented the 3D-image to show that S1P treatment significantly increased intranuclear YAP staining (red) that was colocalized with DAPI (green) (Figure 5A). We have revised the manuscript accordingly (please see lines 251-254 and Figure legend of Figure 5).

P13, L344/345/348 and P15, L425

                                The authors mention “conditioned medium”. To be clear can the authors explain the meaning of “conditioned” in the context?

Response:

Conditioned medium is spent media harvested from cultured cells. We have added the related information in the text (please see line 283).

P15, L415/416    The authors deals with the rapidity of the stimulatory effect of S1P on expression of CCN1 and CCN2 in comparison with COX2 expression. For the latter, the time is not mentioned: to be accurate the time (48h) must be given here.

Response:

We apologize for the confusion and have added the related information (please see the revised Figure legend for Figure 7)

P16, Fig.8 and P15, L432/434

                                As stated in the discussion about the absence of a detailed mechanism of CCN2 mediation of the COX2 expression, Figure 8 should be slightly modify: the arrow linking CCN2 to COX2 should be dotted (and not in plain line) and marked with a question mark. With the actual scheme it seems that CCN2 acts directly on COX2 after exit from the nucleus while, as explain in P15, L435, it should be secreted and acts indirectly on COX2 expresion.

Response:

We agree with the reviewer’s concern and have revised the figure accordingly (please see the revised Figure 8).
